# A computational framework for resolving the microbiome diversity conundrum

**Itay Daybog** [1] ✉ **& Oren Kolodny** [1] ✉

Recent empirical studies offer conflicting findings regarding the relation between host fitness and the composition of its microbiome, a conflict which we term 'the microbial β- diversity conundrum'. The microbiome is crucial for host wellbeing and survival. Surprisingly, different healthy individuals' microbiome compositions, even in the same population, often differ dramatically, contrary to the notion that a vital trait should be highly conserved. Moreover, gnotobiotic individuals exhibit highly deleterious phenotypes, supporting the view that the microbiome is paramount to host fitness. However, the introduction of almost arbitrarily selected microbiota into the system often achieves a significant rescue effect of the deleterious phenotypes. This is true even for microbiota from soil or phylogenetically distant host species, highlighting an apparent paradox. We suggest several solutions to the paradox using a computational framework, simulating the population dynamics of hosts and their microbiomes over multiple generations. The answers invoke factors such as host population size, the specific mode of microbial contribution to host fitness, and typical microbiome richness, offering solutions to the conundrum by highlighting scenarios where even when a host's fitness is determined in full by its microbiome composition, this composition has little effect on the natural selection dynamics of the population.

The microbiome, the diverse community of microbial symbionts associated with a host, can immensely influence its host's wellbeing in numerous direct ways, including providing access to nutrients[1,2], protecting against pathogens[3,4], and inducing resistance to extreme conditions[5,6]. Thus, a variety of traits and diseases have been linked to alterations in microbiome composition and its interaction with the host[7,8], also prompting the issue of its part in ecological and evolutionary processes[9,10]. These have inspired extensive research, particularly in humans, to uncover the mechanisms that govern the host-microbiome interaction. Such studies have been able to link the composition of an individual's microbiome to numerous factors, ranging from the host's health and physiology[11–13], through its behavior[14,15], to its aging dynamics[13,16].

At the same time, studies have also found that the compositions of different individuals' microbiomes, even when considering only healthy individuals in the same population, often differ dramatically[17–19]. Such observations may be baffling and seem to contrast with the mass of evidence linking a host's fitness to the composition of its microbiome. The apparent contradiction stems from the fact that a trait that greatly benefits an individual is expected to spread and fix in the population through selective dynamics, and to be highly conserved[20–22].

This expectation, to find reduced diversity in traits of importance that is driven by selection, is supported by observations across multiple systems and biological modalities: it is widely observed in protein structure[23] and is used to infer aspects of functional importance[24,25]; in genetics, the extent to which a genetic sequence is conserved is commonly used as a measure of its importance for its bearer's fitness[26–28]. Reduced trait diversity in traits of functional importance is also seen outside of biology, as in[29], where the diversity of functional

[1]Department of Ecology, Evolution and Behavior, The A. Silberman Institute of Life Sciences, The Hebrew University of Jerusalem, Jerusalem 9190401, Israel. ✉e-mail: itay.daybog@mail.huji.ac.il; oren.kolodny@mail.huji.ac.il

and symbolic design features of Polynesian canoes was quantified. The existence of standing variation in fitness-influencing traits thus calls for explanation; for example, standing polymorphism in functional traits is often explained as a result of balancing selection[30], frequency-dependent selection[31], character displacement[32], niche heterogeneity[33], or adaptive trade-offs[34]. Driven by the observed variation in the microbiome composition of individuals within the same population, across populations, and among host species, studies have examined the specificity of the relationship between microbes and their hosts. Several such studies focused on gnotobiotic mice and zebrafish, which present highly deleterious phenotypes. It has been shown that introduction of arbitrarily selected microbiota, even from soil or from phylogenetically distant host species, often leads to the successful establishment of microbiota and to significant rescue effects of host phenotypes[35–38]. These findings deepen the conundrum, as they showcase the importance of the microbiome, yet at the same time present the magnitudes of variation it can withstand while still bestowing a fit phenotype. These seemingly conflicting findings pose what we term the microbial β-diversity conundrum.

Although the depth of the conundrum is not frequently appreciated, the surprising diversity of microbiome compositions among healthy individuals in the same population has not gone unnoticed. The most commonly invoked explanation to this puzzle is the possibility that thanks to functional redundancy among microbial taxa, even microbiomes that are taxonomically divergent from one another can have similar functional profiles[17,39,40]. Thus, for example, different microbes may be able to break down a particular complex carbohydrate, and different host individuals can receive this "service" from different microbial species. Although plausible, the empirical support for this claim has been heavily criticized[41,42] and extensive, previously unappreciated, functional diversity among individuals' microbiomes has been reported[41–44].

The paradox can usefully be considered from an eco-evolutionary perspective: on the one hand, the microbiome is able to influence fitness significantly, suggesting that natural selection would shape its composition and that the microbiome would play a role in selection on its host, while on the other hand, it shows a great deal of variation within species, perhaps testifying against its relevance in selection-dictating dynamics and evolutionary shaping of host populations.

Several computational models have been proposed to investigate the evolution of hosts and their microbiome. Most have focused on short-term dynamics, very specific selection-inducing scenarios, or on the population dynamics of the microbiome itself[45–47]. For example, one computational approach has suggested the possibility that selection against toxic stress can drive the co-evolution of adaptive capabilities of the host microbiome[48]. Another model-based study has suggested that to preserve the existence of symbionts that benefit their host at a cost to themselves, very strict conditions must be met[49]. The proposed conditions include fast host reproduction and strong vertical transmission of the microbiome, assertions that have recently been challenged[50].

Despite the rapid developments in the study of the microbiome and the understanding of its importance, research has focused primarily on its influence over short periods on hosts' wellbeing or the evolutionary dynamics of the microbiome alone. Understanding of the dynamics between the microbiome and its associated hosts is still incomplete, and an intuitive and broad theoretical framework that will allow explicit hypothesis testing regarding these dynamics over many generations is still lacking. We attempt to bridge this gap by proposing a framework that implements considerations from the field of microbiome research, alongside perspectives and approaches typically found in studies of population dynamics and evolutionary biology. The framework is designed in a modular and general way, to allow exploration of a broad range of questions on different timescales. Furthermore, this framework can serve as a null model, as it assumes neutral dynamics at the microbial level, in the ecological sense of neutral models[51,52]. In particular, it assumes no niche specialization among the modeled microbes, aiming to adopt the most parsimonious approach and to offer explanations to observed phenomena that rely on as few assumptions as possible.

In this study we use our framework to propose several possible solutions to what we dubbed the microbial β-diversity conundrum, highlighting scenarios in which despite a major influence of the microbiome on each individual's fitness, the role of the microbiome composition on the hosts' population dynamics may vary from great to none. These, in turn, offer testable predictions regarding the conditions in which the microbiome composition is expected to be conserved or divergent among individuals.

## Results
### The model
Our agent-based framework tracks a population of host individuals and their corresponding microbiomes over time. It shares many commonalities with the frameworks that have been put forth by Zeng et al.[45,46]. several differences are discussed in the Methods section. The basic mechanisms of the simulations follow a Wright-Fisher model using discrete non-overlapping generations[53,54]. For simplicity, it is assumed that all hosts reproduce asexually, avoiding the need to explicitly simulate the pairing of individuals for mating. The model focuses on utilizing ecological and evolutionary principles of microbiome transmission, assembly, and contribution to host fitness while setting aside other factors that can be further explored in the future.

We simulate the host as a passive microbe recipient, receiving microbes randomly based on their abundance in the microbial pool available to it. Furthermore, the microbiome assembly is dictated by the arrival order of different microbial taxa, a logistic growth function, and is limited only by the carrying capacity, eliminating effects of any other intra-microbial dynamics, i.e. it is a neutral model in an ecological sense[52,55]. Lastly, after the assembly is complete, there are no further changes in the microbiome configuration of an individual host. Thus, our framework is highly simplified and general, not bound by the explicit mechanisms of any specific host, microbe, or eco-system. This allows it to serve as a tool for understanding underlying patterns that may be difficult to notice or tease apart from other processes in more complex systems, and as a null framework for fundamental host-microbiome dynamics. The framework was designed in a modular way, such that future explorations that apply it may introduce more specific assumptions along any of these dimensions.

The framework can simulate a wide range of eco-evolutionary scenarios. However, the main forward-in-time process remains similar. The host population size $N$ is constant and generations are non-overlapping. To assemble a new generation, $N$ new agents are defined, and a parent is selected from the preceding generation for each one. Next, this individual's microbiome composition is assembled using a microbial pool constructed according to its parental microbiome and the population-wide microbiome. Finally, the hosts' fitnesses are calculated based on the composition of their newly acquired microbiome, thus allowing the generation cycle to continue.

The selection of a parent for each offspring is done by randomly choosing one host from the previous generation, where high-fitness hosts have a proportionally larger probability to be chosen. Each choice is independent, enabling some hosts to produce multiple offspring while maintaining a constant population size (Fig. 1). In a neutral selection scenario, where all host fitnesses are constantly identical, all hosts have an equal probability to be chosen as parents. Thus, in a population of size $N$, random drift will drive a coalescent process of host lineages such that on average all host lineages will coalesce to a single common ancestor after $2N$ generations, following a neutral Wright-Fisher process[53].

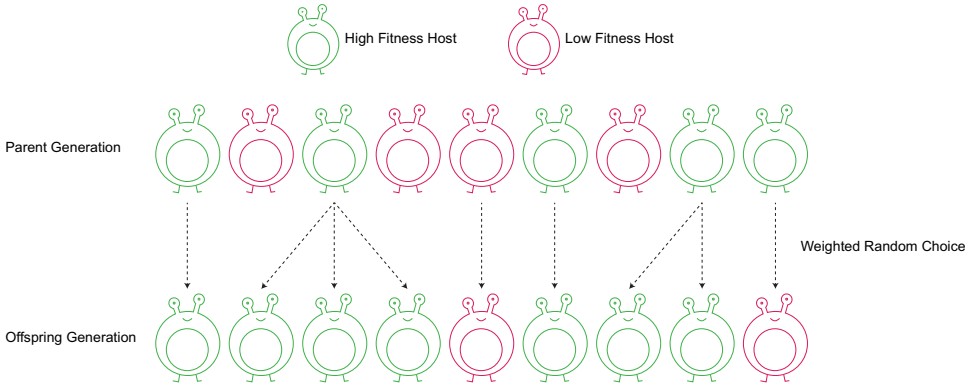

**Fig. 1 | An illustration showing the inter-generational reproduction scheme.** The top row represents the host population of the parental generation, and the bottom row represents the offspring host population. The dashed arrows indicate a parent-offspring relation, and the color represents the host fitness, determined by its microbiome composition (green, high fitness; red, low fitness). The microbiome composition of the offspring is acquired in part from the parent, thus the fitness of parents and their offspring is somewhat correlated. Fitter hosts produce relatively more offspring, and the population size remains constant.

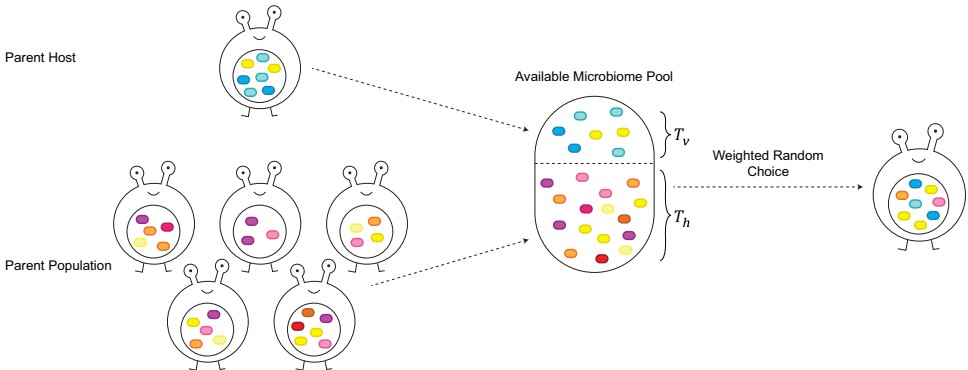

**Fig. 2 | A high-level display of offspring microbiome acquisition from the two available sources.** Parental source (cold hues) and population-wide microbiome (warm hues). Each colored ellipse represents a microbe, where a specific taxon corresponds to a specific color. The two sources dictate the abundance of the microbe species in the microbial species pool from which the offspring will sample microbes, with relation to the vertical and horizontal transmission coefficients $T_v$ and $T_h$.

A host acquires microbes from two main sources. The first is the parental microbiome, contributing to the total microbe pool available to the offspring according to a 'vertical transmission coefficient' $T_v$. The microbiome composition of the parent is normalized to represent the relative abundance of the different microbial species, and later multiplied by $T_v$ to represent its relative contribution to the pool from which the offspring will sample microbes. Thus, if the abundance of a specific microbe species in the parent is $x$, it will contribute $T_v \cdot \frac{x}{\text{all microbes in the parent host}}$ of that species to the microbiome sampling pool available to its offspring. The second source is the population-wide microbiome. This oblique microbiome transmission from non-parental individuals is denoted by the transmission coefficient $T_h$[56]. For simplicity, we refer to this type of transmission as horizontal transmission, in contrast to the parental vertical transmission. Similarly, if the abundance of a microbe taxon within the entire parent population is $y$, the environment will contribute $T_h \cdot \frac{y}{\text{all microbes in the parent population}}$ of that specific taxon to the pool. The ratio between $T_v$ and $T_h$ dictates the transmission scenario being simulated (Fig. 2). For simplicity and increased tractability, to uncover patterns in the microbiome's effect on its host's selection dynamics, we mainly focus in this manuscript on extreme transmission scenarios, where either $T_v = 0$ or $T_h = 0$, corresponding to purely vertical or purely horizontal transmission. Other transmission schemes are feasible:

when $\frac{T_v}{T_h} > 1$, the transmission is mostly vertical, allowing higher conservation of microbiome-related traits between a parent and its offspring, whereas when $\frac{T_v}{T_h} < 1$ the transmission is mostly horizontal, which lowers the correlation between parent and offspring microbiome compositions.

The microbiome assembly process is performed as the available microbe species start inhabiting the host, where more abundant species in the pool of candidate microbes are more probable to be the first to establish within the host (Fig. 2). Between establishment events, all previously acquired microbe populations grow logistically, limited by a predefined maximal microbial species' population size. The host's microbiome takes shape as more species join and grow in number until the overall carrying capacity of microbes in the host is reached.

A host's fitness score is calculated by summing the individual contribution of each microbe taxon it possesses. For simplicity, the calculation in the simulations we carried out was done based on the presence/absence of each microbial species, regardless of its abundance. This may occur, for example, if the helper microbiome supplies a vital nutrient, otherwise inaccessible to the host, required in small amounts[57]. Each microbial species contributes a certain value to a host's fitness score. At the start of each simulation, each microbial species is randomly assigned a fitness value that this species' presence will contribute to the host, drawn from a distribution (Fig. 3). Several contribution distributions are plausible, including a step distribution, where some taxa contribute much while others contribute little (Fig. S1a),

**Fig. 3 | An example of the fitness calculation of a single host.** The contribution of each microbe taxon (left) is predetermined at the start of each simulation according to various parameters. The fitness of each host is calculated (right) according to it and to the microbe taxa it possesses. The calculation does not address taxa abundance but rather acts as a linear sum of the contributions of species present in the microbiome. After the fitness of the entire population is calculated, it is divided by the maximal value to adhere to the traditional [0,1] fitness range.

a low-variance distribution, determining that the contributions of different taxa are similar, and an almost uniform distribution, leading to a broad range of different contributions by different taxa (Fig. S1b).

The simulation continues until the number of common ancestors of the hosts in the population, denoted as AC for 'ancestral coalescence', reaches a predetermined value. For example, to follow the simulation until all the hosts in share the same common ancestor, the simulation is run until AC = 1.

We used the framework to execute simulations under various combinations of parameters. These include different microbial assembly and transmission factors, varying host population sizes, and a few other ecological components of the model. With these we were able to detect and explore scenarios that may hold the solution to the conundrum highlighted above, highlighting situations in which the microbiome determines host fitness while remaining relatively non-conserved.

### Running and interpreting simulations

The β-diversity conundrum arises from empirically supported and seemingly contradicting observations—on the one hand the microbiome is crucial for host fitness, while on the other hand it can differ vastly even among healthy individuals in the same population. The contradiction stems from the latter being an unexpected characteristic for a fitness-determining trait, which are typically highly conserved. We seek solutions to the conundrum in the form of scenarios in which both observations are true and their co-existence is interpretable. For this, we consider the most conservative scenarios with respect to the first of the two conditions we are after. Firstly, in our simulations the host fitness is solely a function of microbiome composition, epitomizing the dependence of the fitness of hosts on their microbial symbionts. Secondly, microbiome diversity among individuals would arise when selection cannot effectively act to favor one composition over another via selection on hosts. An obvious setting in which this would occur is when microbiome composition is not heritable. This trivial case of purely horizontal transmission of microbes is explored in the supplementary. We focus instead on the setting at the other end of the spectrum: the case in which transmission is purely vertical. Solving the conundrum is thus reduced to detecting in our framework, under these most conservative conditions, fundamentally different scenarios in which selection on microbiome composition is ineffective.

### Measures of microbiome influence on natural selection

We used two measures to evaluate the microbiome's effect on selection dynamics in the host population. The first is the observed difference between the fitness scores of hosts with different microbiome compositions, expressed in the distribution of host fitnesses in a single generation. This acts as a direct intragenerational approximation of how the microbiome influences the differential of selection among individuals and is not directly influenced by different microbial transmission schemes. The second measure is the number of generations that pass until AC reaches a predefined value. Taken from the field of population genetics, this measure quantifies the long-term effects of the microbiome on the selection dynamics in the host population over many generations.

Applying the two measures, we characterized conditions under which the microbiome can substantially influence host selection, alongside key scenarios where different microbiome compositions have little influence or none, even though the microbiome in our framework is the sole contributor to the fitness of its host.

### Species-rich microbial configurations facilitate high diversity among hosts of microbiome compositions that all lead to similar fitness values

We first examined the possible effects of the microbiome's α-diversity, microbiome diversity within individuals, on the selection-related dynamics of the host population, by simulating two microbiome structures. The first structure represents a species-rich microbiome composition, both in the number of species and their abundance, corresponding to the one that is characteristic of many vertebrate host species: each individual carries a microbiome composed of between 200 and 300 microbial species, ranging in relative abundance from highly prevalent to rare, with few particularly common taxa accounting for the majority of the overall microbial biomass of that individual (see, e.g.[17,58–60]) (Fig. S2). The second structure is the complement of the latter, exhibiting a species-poor composition with a lower number of species and a lower carrying capacity. This structure simulates the microbial composition found for example in many insects, where typically one or two microbial species dominate and only a few others are sparsely present[61,62] (Fig. S2b). In the supplementary we explore several additional compositions (Fig. S3 and respective results in Figs. S8–S9).

We ran 100 repeats of the simulation with each of the two host population types—the species-rich and species-poor microbiome configurations under vertical microbial transmission, where each microbe species contributes differently to its host fitness. The results display the empirical distribution of host fitness scores observed in the first generation (Fig. 4a). The fitness scores of the hosts in the species-poor population are widely distributed and show great variance (Fig. 4b), underlining that some hosts have gained a very high relative fitness score by acquiring the most beneficial microbes, whereas others did not.

In contrast, we find that the host fitness scores in the species-rich population present lower variance ($p < .001$) (Fig. 4b). This results in the least fit host being almost equivalent to the fittest one, stripping the microbiome of the ability to influence host selection. Notably, both in the species-rich and in the species-poor microbiome configuration

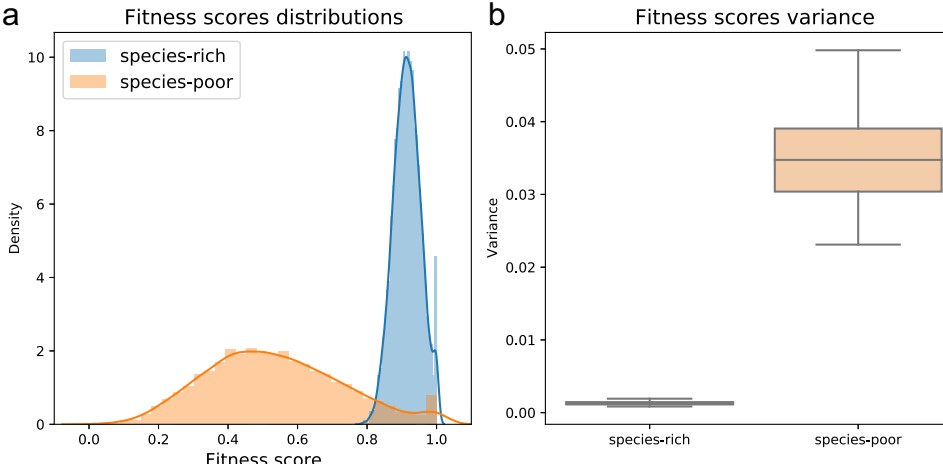

**Fig. 4 | The influence of the microbiome's species-richness on first-generation fitness scores across 100 repetitions.** This is shown for both scenarios: a species-rich microbiome (blue) and a species-poor microbiome (orange). **a** Normalized histograms representing the distribution of the host population's fitness scores. Lines represent the Gaussian kernel density estimation of the fitness scores distribution. Each scenario shows 5000 individual hosts. **b** boxplots displaying the observed fitness scores variances across 100 repetions of the simulation. Boxplots here and throughout the manuscript are presented with the sample median and a box representing the 25th to 75th percentiles. Whiskers portray the sample minima and maxima. Here and throughout the manuscript, double-sided $t$-tests were used for calculation of statistical differences between groups.

scenarios the $\beta$-diversity within the population at the beginning of each simulation is high, with mean pairwise Jaccard distances between individuals of 0.82 and 1 respectively. In these simulations, $\beta$-diversity changes over time; only in the species-poor scenario do fitness differences among hosts drive selection, leading the lineage with the beneficial microbiome to spread and to a respectively rapid decrease in the population's mean $\beta$-diversity (see also supplementary figures S12-S13). Combining these findings, we conclude that a species-rich microbiome configuration may act as the first key to the microbial $\beta$-diversity conundrum. It presents a state where the microbiome affects fitness and is diverse between hosts, yet it does not create a noticeable fitness difference, giving rise to neutral population dynamics.

These results can be attributed to the law of large numbers, which formulates the tendency of large sample sizes to approximate well the mean of a hidden distribution[63]. We simulate the microbial species' different contributions to host fitness such that they can be viewed as discreetly drawn random variables from some background distribution. Thus, when summing the fitness contributions of species in a microbiome configuration, we expect to get an approximation of the mean of the background distribution multiplied by the number of species in that microbiome. According to the law of large numbers, when the microbiome is species-rich the approximation of the multiplied mean in each host will be much better than when the microbiome is species-poor. This leads to the low variance in fitness scores observed between hosts in populations with species-rich microbiome configurations, creating a situation in which the microbiome is unlikely to generate a large enough fitness difference between hosts to significantly influence their selection dynamics.

This scenario, in which very different microbiomes lead to similar overall contribution to host fitness, shares features with the commonly invoked solution to the $\beta$-diversity conundrum, which was mentioned earlier: that different microbiome compositions, thanks to functional redundancy among different species, share functional similarities that allow them to provide the same "services" to the host[17,40,64–66]. In ecological terms, this explanation relies on a niche-based perspective, assuming that functional niches exist in the gut and may be filled by a range of microbial species. The solution proposed here based on our simulations does not contradict this niche-based explanation, and in fact aligns well with it. However, it relies on less assumptions. As our

framework is neutral (in the ecological sense, with respect to microbial dynamics and functions), it is more parsimonious and sidesteps the criticism that has been levelled at the functional redundancy hypothesis (e.g[41,42]).

To validate our hypothesis on a longer time scale, we address our second measure–the number of generations to ancestral coalescence under three microbial transmission schemes–pure vertical, pure horizontal, and an equal combination of the two, dubbed 'midway' transmission. We compare the number of generations until coalescence in each population to this measure in a neutral scenario where the coalescent dynamics are only the product of random drift, and the microbiome is irrelevant to the hosts' fitness. Under vertical transmission, The simulations' results show that the time it takes for the populations with the species-rich microbiome to coalesce is almost identical to the neutral scenario ($p = 0.99$), while the populations with the species-poor microbiome composition coalesce to a single lineage in half that time on average ($p < 0.001$) (Fig. 5a). This indeed matches our results using the first measure, further implying that the microbiome is not affecting the hosts' natural selection when it is highly $\alpha$-diverse, and vice versa.

When addressing pure horizontal transmission dynamics, the coalescence times of both the species-rich and species-poor microbiome configurations behave similarly to that of the neutral dynamics (Fig. 5c). This is the expected outcome, as when the microbial configuration is not at all linked to the ancestry of the individual host, it will not be able to benefit specific lineages, leaving the assembly of the microbial configurations in each generation to neutral processes. Under the midway microbial transmission scheme (Fig. 5b), the species-rich configuration remains indistinguishable from neutral dynamics ($p = 0.81$), but the coalescence times of the species-poor populations are faster relative to it and to the neutral dynamics ($p < 0.05$), yet not as fast as in purely vertical transmission. This coheres to the nature of the "midway" transmission as a mixture of both vertical and horizontal transmissions–the vertical transmission enables host lineages to acquire unique microbial compositions that bestow different fitness scores, thus driving the system to faster coalescence, while the horizontal transmission somewhat breaks this exclusivity, delaying coalescence (Fig. 5c).

The results in the species-rich populations showcase a feasible scenario where both sides of the conundrum are true

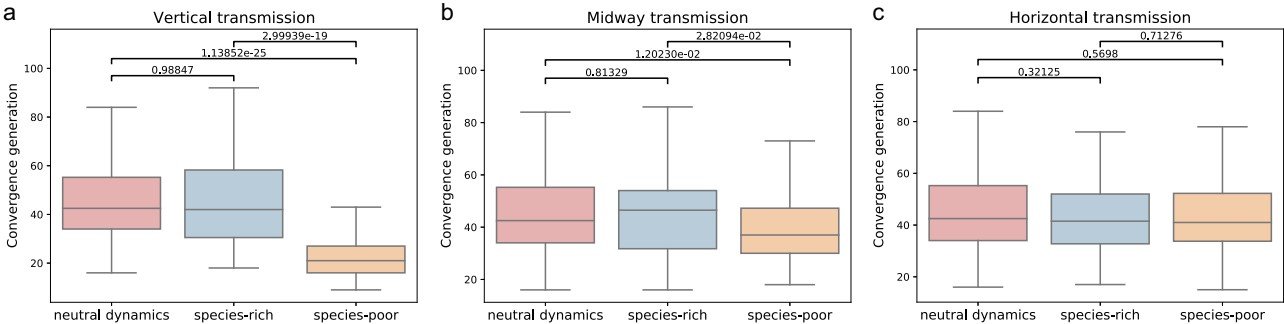

Fig. 5 | Influence of the microbiomes' species richness on the number of generations it took for all existing hosts in the population to share a common ancestor. Results are calculated across 100 repetitions of the stochastic simulation. Neutral scenario without microbiome effect (red), species-rich microbiome (blue) and species-poor microbiome (orange). **a** Under purely vertical transmission. **b** Under mixed transmission (half vertical, half horizontal). **c** Under purely horizontal transmission.

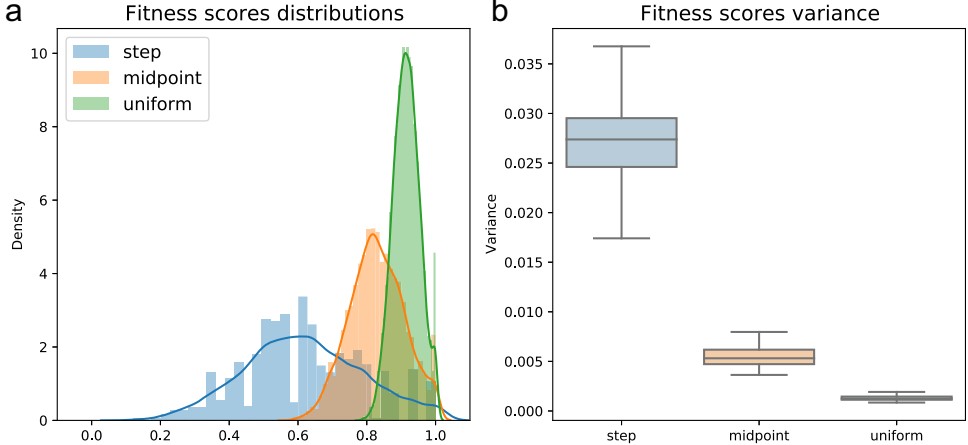

Fig. 6 | Influence of different distributions of microbes' contributions on host fitness. These are shown in a species-rich host, for first-generation fitness scores across 100 repetitions for three contributions' distribution scenarios: uniform (green), step (blue) and a midpoint between the two (orange). **a** Distributions of fitness scores. **b** Variance in fitness scores. See also Fig. 4.

simultaneously—The microbiome is the sole contributor to the fitness of its host, yet still it is not able to influence the selection dynamics in the host population. This observation, under the most extreme case of purely vertical microbial transmission, highlights the richness of the microbiome compositions as a possible solution to the microbial $\beta$-diversity conundrum.

### Large differences in the contributions of microbial species to fitness lead to effective selection among hosts

We saw that species-rich microbiomes are less likely to drive selection. So—is microbiome composition irrelevant to selection in species that typically have high $\alpha$-diversity microbiomes? We tried searched for conditions which would invalidate the assumptions of the law of large numbers. This would require that although many species are present in the microbiome, the sum of their contributions still would not approximate the multiplied mean of the background contributions distribution well. Thus, we tested the influence of altering the distribution itself, i.e., the contributions of microbe species to the fitness of their hosts. We examined three background distributions of contributions-to-fitness among the different microbe species: The first is a step distribution, where each microbe species contributes either the maximal or minimal value possible, a trait set only once at the beginning of the simulation by random sampling with the probabilities 0.025 and 0.975 respectively (Fig. S4a). This Effectively leads to 2.5% of the microbiome taxa to greatly contribute to the host fitness, while the

rest contribute very little. The second is an almost uniform distribution where each contribution value between the minimal and the maximal values is equally represented, as was used in the previous section (Fig. S4b). The third is a midpoint between the two previously described schools, where most species would result in contributing little while few species contribute greatly (Fig. S4c).

We ran the simulation under the three scenarios on populations of hosts with the species-rich microbiome configuration. As before, we first look at the distribution of fitness scores in the first generations, to understand the microbiome's predisposition to influence host selection under each scenario. Truly, we see that altering the scheme by which the microbiome contributes to its host's fitness has an impact on the distribution of fitness scores (Fig. 6a). When comparing the three scenarios, the observed trend is of an increase in fitness scores' variance as the microbiome contribution distribution is less uniform ($p < 0.001$) (Fig. 6b).

These findings correspond to the law of large numbers: when the distribution's variance is larger, a greater sample size is required to approximate its mean[63,67]. Thus, when the contribution of each microbial species to its host's fitness is drawn from a high-variance distribution, even in species-rich microbiomes the number of species may still be small enough such that different microbiome compositions lead to significantly different fitness scores.

The long-term effect of the different microbial contribution distributions, as seen in the time until coalescence under the vertical

transmission scenario, also supports this hypothesis. We indeed see that under the uniform-distribution scenario the times to coalescence are quite similar to the ones under neutral dynamics ($p = 0.99$), meaning the microbiome did not have a significant influence on population dynamics of the hosts (Fig. 7). In contrast, when the variance in contributions is large, the time to coalescence shortens by half ($p < 0.001$), indicating that the microbiome did take part in driving the host selection processes by allowing the more fit host lineages to take over the population within a smaller number of generations.

Under purely horizontal transmission dynamics, we see that the time to lineage coalescence remains similar to that of the neutral dynamics under all the three different distributions of microbial contribution ($p > 0.3$) (Fig. S6a). This is reasonable as neutral selection dynamics are expected when the fitness of hosts is not strongly linked to their ancestry, and the fitness is determined by many components, reducing the effect of small fluctuations in its composition. The results are similar under a midway microbial transmission scheme (Fig. S6b).

The results arising from these simulations, especially under purely vertical transmission, show that in host populations with species-rich microbial compositions a possible solution to the conundrum could lie

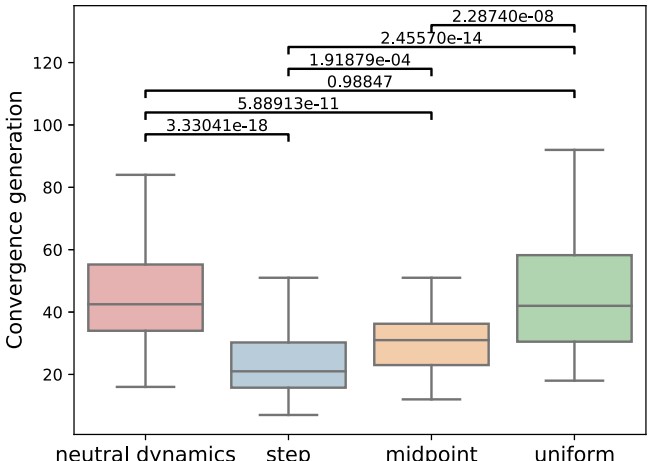

**Fig. 7 | Influence of different distributions of microbes' contributions to host fitness.** Explored in species-rich hosts, shown via the distributions' influence on the number of generations it took for all existing hosts in the population to share a common ancestor, under purely vertical transmission, across 100 repetitions of the stochastic simulation. Neutral scenario without microbiome effect (red), uniform (green), step (blue) and a midpoint between the two (orange).

within the particular fashion in which the microbes contribute to the fitness of their hosts. For example, if the distribution of the microbial contributions to the fitness of the hosts is uniform, then the microbiome does not affect the selections dynamics of its hosts despite being the sole determiner of their fitness.

## Microbiome is more likely to drive selection in large host populations

Another theoretical and empirical factor known to play a prominent role in the population dynamics and its selection dynamics is its size[68–72]. We thus set out to test whether the size of the host population can play a part in the microbiome's ability to drive natural selection dynamics among its hosts. To do so, we simulated host populations varying only in the magnitude of the number of hosts that comprise them: 20, 200, and 2000 hosts. The hosts' microbiomes were simulated using the species-rich configuration, and the distribution of microbes' contribution to their host's fitness was under an almost uniform scenario (see Fig. S10 for a complementary exploration of population size effect in hosts with species-poor microbiomes).

We begin by looking at our short-term indicator for the microbiome's ability to influence host selection, the distribution of fitness scores in the first generation (Fig. 8a). We see that unlike the factors that we tested previously, the population's size does not lead to large differences in the distribution or variance of the hosts' fitness scores (Fig. 8b). In other words, the differential of selection among host lineages is small in this case and is not affected by population size.

Perhaps surprisingly, however, when we examine the multi-generational effects of the population size on the microbiome's influence on selection, we find that population size alters the extent to which the microbiome influences population dynamics. To compare coalescence times in populations of different sizes under purely vertical transmission dynamics, we normalize the generation of coalescence by the population size, $N$ (Fig. 9). We see that the larger the population, the relative number of generations needed to reach a state where all hosts share a common ancestor becomes significantly shorter ($p < 0.001$),. This suggests that although a single-generation indicator does not show, for different $N$ values, a different differential-of-selection among lineages, the microbiome is more capable of driving selection in larger host populations. This is due to the increased efficacy of selection in large populations, and the increased likelihood that in a large population, even small fitness differences will be realized, as is well-known in models of genetic evolution[69–71,73,74]. This contrasts with the dynamics in small populations, in which random drift is a relatively more prominent force[69–71,73], and in which effective selection due to microbiome-mediated fitness effects seems

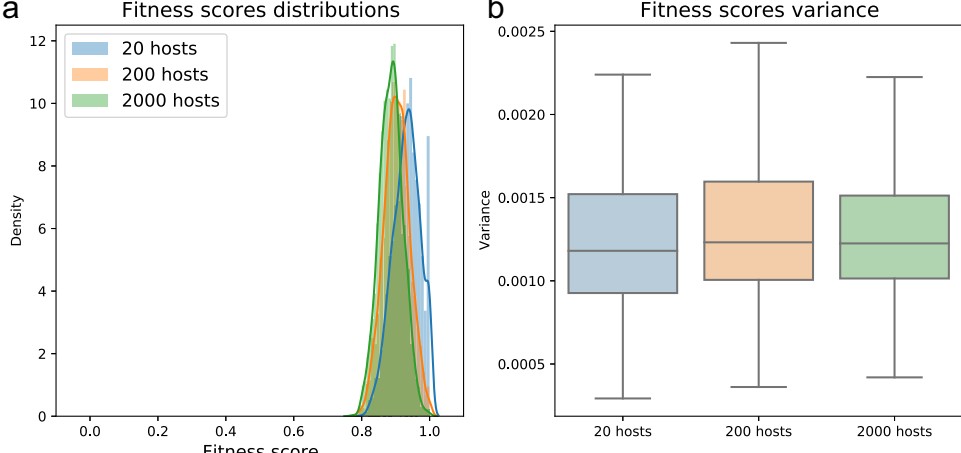

**Fig. 8 | Influence of host population size on first-generation fitness scores.** Shown across 100 x $n$ hosts. $n$ is the number of hosts in the smallest population size scenario, $n = 20$. Shown for three size scenarios—20 (blue), 200 (orange) and 2000 (green). **a** Distributions of fitness scores. **b** Variance in fitness scores. See also Fig. 4.

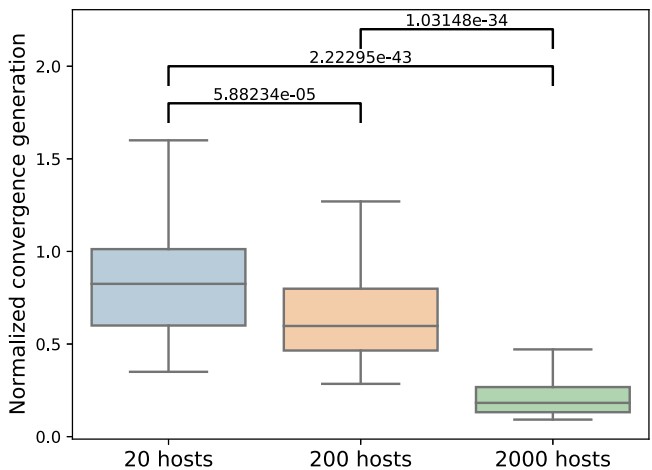

**Fig. 9 | Influence of host population size on the number of generations it took for all existing hosts in the population to share a common ancestor divided by the populations size.** Shown under purely vertical transmission in the species-rich microbiome scenario, across 100 repetitions of the stochastic simulation. 20 hosts (blue), 200 hosts (orange) and 2000 hosts (green).

less likely. As expected, under horizontal and midway microbial transmission schemes, the size of the host population does not significantly affect the time to lineage coalescence (Fig. S7). We thus find that, in line with classic population genetics' theory, in smaller host populations the microbiome is limited in the extent to which it can influence the selection dynamics of its hosts. This highlights the population size as another solution to the seemingly conflicting aspects of the microbial β-diversity conundrum.

## Discussion

In this paper we highlight an overlooked conflict in empirical findings from microbiome research—'the microbial β- diversity conundrum'—and attempt to reconcile it by introducing a simple and modular framework capable of simulating the evolutionary and ecological dynamics of a host population and their associated microbiomes. We propose different answers to the paradox by simulating various scenarios, including different assembly and contribution dynamics of the microbiome, different microbial transmission schemes, and different host population-related parameters. Our method of suggesting solutions to the puzzle was to demonstrate probable scenarios where the microbiome alone is affecting its host fitness while also displaying high β-diversity and inability to drive selection between the hosts in the population. In these scenarios, we aim to also pinpoint the parameters that facilitate this duality. In this article we present three such scenarios—a species-rich microbiome configuration, a relatively uniform distribution of contributions to host fitness between microbial species, and a small host population size.

Resolving the conundrum means settling the conflict between the empirical findings which brought the microbial β-diversity conundrum to light— ones that underline the microbiome's importance to its host's fitness on one hand[1–8,11,13–16,75], but that do not lead to the conservation of a particular microbiome structure in a population[17,18]. We find that one solution may lay in the composite nature of the microbiome. Being composed of many different species with different contributions, as opposed to traditional traits which are usually thought of when discussing trait conservation, the overall influence of the microbiome on its host is subjugated to the law of large numbers. As such, we expect that species-poor microbiome configurations are more likely to be driving selection among hosts, and thus be respectively more conserved. This hypothesis is supported by empirical observations in insects, which are characterized by microbiomes composed of

relatively few species, where greater uniformity in microbiome compositions has been reported[76–80]. Such a difference in the compositionality of the microbiome may also occur between different body sites' microbiomes in the same host; in humans, for example, the vaginal microbiome is relatively species-poor, and—in line with the hypothesis above—is characterized by relatively low within-population β-diversity[81,82].

Our findings may direct further research to empirically validate whether the microbial diversity conundrum can truly be explained by the solutions we have suggested here. Novel research can focus on the real-world causes in implications of the conundrum, and test if more microbiome β-diversity or conservation is found in populations that are characterized by one or more of the factors that were discussed, per the results of our simulations.

Notably, the three factors that we highlight as providing possible solutions to the conundrum do so in qualitatively different ways, with rather different characteristics. All three factors: microbiome diversity, contribution distribution of the microbiome, and host population size can influence the number of generations needed for the host lineages to coalesce, yet only the first two impact the fitness scores' distributions in the first generation. This difference underscores the multiplicity of types of factors that take part in the correlation between microbiome and host selection. The solutions regarding the microbiome richness and distribution of contributions are driven by the tendency of selection to be subject to random sampling and regression towards a mean, with results that are mediated by the law of large numbers and are thus evident even in a single generation. In contrast, the solution related to the host population size influences selection dynamics over multiple generations, creating (or not creating) slight selection biases very gradually, which are not necessarily noticeable within a single generation. Interestingly, these factors require different perspectives for their consideration and are traditionally treated within different sub-disciplines of academic study; they are thus rarely considered within the same framework.

This also highlights a significant difference between our two measures of the microbiome's influence on host natural selection. The first, the distribution of fitness scores in the first generation, corresponds to what is typically monitored in empirical studies of microbiome and its effect on its host's wellbeing. Discussing the variance of fitness scores in our simulation parallels with examining the phenotypic diversity among hosts with varying microbiome compositions, which is what initially sparked the notion of the microbiome's importance. In contrast, the second measure—the time to ancestral coalescence—parallels the microbiome to a hereditary trait. By summarizing its impact on selection processes through multiple generations this measure may be more appropriate for inference or exploration of long-term evolutionary dynamics of the host population, closer in spirit to frameworks of population genetics and molecular evolution[9].

To keep our framework general and modular, most of the 'real-world' processes regarding the population and ecological dynamics of the microbes and the hosts were not implemented. Hosts reproduce asexually and are passive in the microbiome acquisition process, the microbes do not interact among themselves in ways other than being subjected to the host's carrying capacity, and after the initial assembly process ends, the microbiome remains constant throughout the host's life span. The framework was designed in a modular way, allowing future incorporation of these processes, alongside other dynamics that were not implemented in the model. By using a simplified model, we were able to identify underlying factors mediating the microbiome's ability to affect host selection even under simplified and selectively neutral intra-microbiome dynamics. The determinants of the microbiome's β-diversity in reality may be many; furthermore, even within the simple version of our framework that was used in this study, a broad range of values of β-diversity can occur, depending on the parameter values used for the microbiomes' assembly; here, we

focused on highlighting scenarios in which high $\beta$-diversity would be maintained over evolutionary time-scales despite a significant role of the microbiome in determination of fitness. In the future, our framework may also be utilized to explore the expected $\beta$-diversity in more specific scenarios, such as for a certain species in which the real parameter values are known. We have also focused here on the hardest-to-explain scenario with respect to $\beta$-diversity, namely the case in which it fully determines the host's fitness and yet remains high. Our framework can also be used to explore scenarios in which $\beta$-diversity is low in the first place due to the assembly process or other constraints, or to study cases in which fitness is only partially dependent on the microbiome's composition.

The framework's modularity enables it to simulate a broad range of scenarios, and may be used in future research of questions that are unrelated to the microbial $\beta$-diversity conundrum. For example, the simulations can be used to predict the environmental rescue effect of microbial species within the microbiomes of a host population. In other words, it can be used to analyze the probability, and the factors controlling it, of species of microbes that were extinct in a population's microbiome to spread once again through the population if reintroduced at some rate from the environment. This can contribute to a current ongoing discussion about the possibility and benefits of human microbiome rewilding—the act of reintroducing lost microbe species to human microbiomes to regain health benefits our hunter-gatherer ancestors possessed[83,84].

In conclusion, we have focused on a surprising paradox that has gone largely unnoticed thus far—the microbial $\beta$-diversity conundrum, an apparent conflict between two commonly discussed findings regarding the microbiome. We have attempted to understand how the composition of the microbiome can be crucial for host fitness, while also being highly divergent among healthy individuals. Using a series of simulations, our research presents a list of several probable factors that could enable this duality—a species-rich microbiome composition, a uniform distribution of microbial contributions to host fitness, or a large population size of hosts. Not only can these solutions resolve the paradox, but also direct further research regarding the microbiome's diversity and the intricate relationship between hosts and their associated microbes. Furthermore, the presented framework is modular and can be used to explore a range of additional topics in microbiome research.

## Methods

The framework implements an agent-based simulation of a host population consisting of a fixed number of individuals, $N = 50$, with $B = 2000$ microbe taxa available in the environment unless stated otherwise. Each executed simulation was run with AC = 2, meaning until the population was comprised of hosts sharing no more than two common ancestors.

### Measuring the microbiome's influence on host selection

The first measure we used to study selection dynamics—the distribution of host fitness scores in a single generation—is directly derived from the individuals' microbiome compositions and correlates with each individual's expected mean number of offspring, thus acting as a relevant indicator for the microbiome's effect on host population dynamics. We approximated the magnitude of the difference using the fitness scores' distribution's variance, comparing it for the first generation under each simulated scenario. Although it is a direct measure, it is also a short-term one, appliable only for individual generations.

The second measure of selection dynamics is the relative time it takes for all the hosts in the population to share the same ancestor. This evaluation is taken from the field of population genetics and is an adaptation of the time it takes an advantageous allele to fixate in a population, paralleling the alleles with the microbiome[71]. Naturally,

it is most relevant when microbiome transmission is mostly vertical since an influence on the coalescence generation is expected only to a limited extent in scenarios where the microbiome is not strongly correlated to specific lineages, as is the case in horizontal transmission.

### Generation of microbiome templates

Instead of real-time calculation of each host's exact microbiome structures, "empty" microbiome templates denoting only the number of species in the microbiome and their abundances were pre-generated, only to be assigned specific taxa during the simulation itself. In the moment of creation, a host is assigned such a template—empty slots varying in size representing abundance, each to later be allocated to a different microbe taxon, out of a total $B$ existing taxa. The microbiome templates are generated by simulating microbial establishment events, where each represents one slot in the final template. As these consecutive events act as a Poisson process, each waiting time $t_i$, between establishment events $e_{i-1}$ and $e_i$, is drawn from an exponential distribution with a tunable rate parameter $\lambda_1$:

$$\forall i \in \mathbb{N} \; t_i \sim \text{Exp}(\lambda_1) \tag{1}$$

Afterward, each waiting time is multiplied by an establishment probability coefficient $s_i$, whose value reflects the probability of a successful establishment event, as a function of its chronological order. In our framework, three such coefficient vectors can be applied, each describing a different microbiome acquisition scenario. (i) A null scenario where all establishment probabilities are 1, thus $t_i$ remains as is (Fig. S5a). (ii) The earlier the establishment event, the more likely it is, simulating the growing struggle for space and resources when more and more taxa inhabit the same niche[85]. The scaling factors follow a scaled exponential decay with a scaling factor $E_s$ and a rate $\lambda_2$ (Fig. S5b). (iii) The highest establishment probabilities are received after several pioneer taxa have already established within the host, colonizing it, and making it more habitable[86], followed by a decrease in the probability, like in the previous case. This creates a "hump" shaped probability vector, with parameters $a,b,c$ creating the hump parabola $ax^2 + bx + c$, alongside $p$ controlling the index of the event with the highest priority and a scaling factor $H_s$. The vector is then normalized to the range $[0 \ldots 1]$, and the minimal probability is set to $H_m$ (Fig. S5c). The explicit notation of the three establishment probability vectors is:

$$\{1 \mid i \in \mathbb{N}\} \tag{2}$$

$$\left\{ \frac{E_s + e^{-\lambda_2 i}}{1 + E_s} \mid i \in \mathbb{N} \right\} \tag{3}$$

$$\begin{aligned} &Let \; P = a(i-p)^2 + b(i-p) + c \rightarrow \\ &\left\{ \max\left( \frac{1}{1+H_s} \cdot \left( H_s + \frac{P - \min(P)}{\max(P) - \min(P)} \right), H_m \right) \mid i \in \mathbb{N} \right\} \end{aligned} \tag{4}$$

The drawn waiting times are scaled by the scaling vector, relatively shortening the waiting time of a high probability event and elongating that of a low probability one. Thus, the final waiting times, $t_{f_i}$, are:

$$\forall i \in \mathbb{N} \; t_{f_i} = \frac{t_i}{s_i} \tag{5}$$

During these waiting times, the already established taxa grow in number in each timestep according to a classic logistic growth function, where $C_s, k, m$ represent the maximal size of a single taxon within the host, the growth steepness, and the sigmoid midpoint respectively. The population size of a single microbe taxon after

$t_i$ time would be:

$$\frac{C_s}{1 + e^{-k(t_i - m)}} \tag{6}$$

The microbiome template's computation is finished when the sum of the abundances of all microbe taxa within the host has reached a predefined global capacity, $C_g$.

Different parameters were used to pre-generate 10,000 different microbiome templates for both the species-rich and species-poor microbiome configurations, to be chosen from randomly during the initialization of each host in the simulations (Tables S1a, S1b). To reach species-rich microbiome templates, waiting times were scaled according to scenario (iii), and to reach the species-poor templates, waiting times were scaled according to scenario (ii). Supplementary figure S3 depicts microbiome templates that were constructed with slightly different parameters, used in simulations that are described in the supplementary material as well.

### Microbiome acquisition

After a microbiome template is assigned to the host, the acquisition process of microbiome equals to assigning each empty slot a unique microbe taxon, denoting its abundance within the host. The first generation of hosts in each simulates being randomly seeded with different microbes out of the possible $B$ taxa. Each host randomly selects microbe species according to the number of slots in its microbiome templates, and randomly assigns each taxon to a different slot thus creating a diverse initial host population.

In the next generations, the host acquires the microbiome through randomly sampling available microbes from a distribution dictated by its parent and the entire previous population. The parental source, $P$, is simply the microbiome of the host's parent normalized to represent the relative abundance of its microbes, and the population-wide microbiome, $E$, is the per-taxon summation of abundances in the previous population's microbiomes, also normalized. The final available microbe distributions a summation of the two sources, weighted by the vertical and horizontal transmission coefficients $T_v, T_h \in \mathbb{R}^+$, representing the relative contribution of the parental source and population-wide source respectively:

$$\text{abundance of taxon i in the sampling distribution} = T_v \cdot P_i + T_h \cdot E_i \tag{7}$$

The microbes from the available pool are randomly assigned to the slots, weighted by their abundance in the pool. Thus, taxa that are more abundant in the joint contribution of the sources are more likely to establish first and inhabit larger slots in the host microbiome's template.

### Microbiome contribution

The specific contribution of each microbe species to the host's fitness is generated at the start of the simulation and remains constant throughout. For each taxon, a contribution value is randomly selected within the range of $C_{min}$ and $C_{max}$, representing the minimal and maximal possible contributions respectively. The random sampling is altered by various parameters following the simulated scenario and is factored by the contribution rate parameter $\lambda_3$ (Table S2). (i) A "step" contribution scenario, where some taxa contribute $C_{min}$, others $C_{max}$, and $\lambda_3$ represents the number of taxa to contribute $C_{max}$ (Fig. S1a). (ii) An "exponential decay" contribution scenario, where the sampling is weighted according to an exponential distribution $Exp(\lambda_3)$. Large $\lambda_3$ values denote a steep density distribution, eventually resulting in a small number of microbes that contribute a lot, while the rest barely do. Whereas small $\lambda_3$ values denote an almost uniform contribution distribution (Fig. S1b).

### Fitness calculation

The fitness of a host is a linear summation of all the contributions of microbe taxa that dwell in its microbiome, summing a species contribution only once if it is present in the microbiome, dismissing its abundance. Meaning for host $i$, the fitness - $f_i$, is calculated as follows:

$$f_i = \sum_{j=1}^{B} 1_{\{taxon\, j\, in\, host\, i's\, microbiome\}} \cdot c_j \tag{8}$$

After the fitness of the entire population is calculated, it is divided by the maximal value in order to adhere to the traditional [0,1] fitness range.

### Jaccard distance calculation

Jaccard distance between the microbiome configuration of two hosts $h_A$ and $h_B$ with microbiomes $A$ and $B$ was calculated traditionally:

$$\text{Jaccard distance}(h_A, h_B) = 1 - \frac{|A \cap B|}{|A \cup B|} \tag{9}$$

The representing Jaccard distance of a population $P$ was calculated as the mean of all Jaccard distances within every two hosts in the population:

$$\text{Jaccard}(P) = \frac{\sum_{i,j \in [1...N], i \neq j} \text{Jaccard distance}(h_i, h_j)}{\binom{N}{2}} \tag{10}$$

### Statistical significance calculation

We used standard two-sided $t$-tests to determine whether observed differences between different groups were significant. Wherever first generation fitness scores were compared, the compared groups were composed of 100 repetitions $\times N$ hosts. Where $N$ was different across comparisons the calculations were done on the minimal $N$. Each set of compared times to coalescence was comprised of measurements from 100 stochastic simulation repetitions.

### Boxplot information

All boxplots in this manuscript are presented with the sample median and the box representing the 25th to 75th percentiles. Whiskers portray the sample minima and maxima.

### Relation to other generative frameworks for modeling host-microbiome dynamics

Several computational frameworks have been constructed so far for the modeling of host-microbiome dynamics in population contexts of both hosts and microbes[45–47,49,50,87,88]. Among these, the framework we propose here is most similar to the two models proposed by Zeng et al.[45,46]. Our implementation is independent of theirs, and a detailed comparison among them in the future may be productive, as it may highlight qualitative differences that arise from seemingly arbitrary modeling choices and implementation practices. Two particularly notable differences between Zeng et al.'s (2015, 2017) frameworks and our framework have to do with the dynamics of construction of a host's microbiome, and the way in which the microbiome may influence host fitness.

(1) A host individual's microbiome in the Zeng et al. framework (2015) is composed of microbes that occupy a pre-determined number of slots ($n = 1000$, or $n = 100,000$, for example); these slots are filled via multinomial sampling from several possible sources according to simulation parameters. Our framework's scheme of populating the microbiome is slightly more ecologically realistic: we distinguish between transmission of microbes (a rare event, of one individual microbe) and their establishment (which may

depend on the number of previously established species, for example, even in a niche-neutral model, as described above) and multiplication within the host, from a single individual to $10^3$–$10^8$ individuals, depending on how early in the colonization process the microbe arrived. Depending on model parameters these processes can lead to a range of microbiome structures as discussed above, and the model can thus be used to explore the ecological determinants of microbiome assembly and their interaction with dynamics on evolutionary timescales (of multiple host generations), including incorporation of a range of ecological considerations and their influence on patterns of microbial diversity. This includes, for example, direct comparison between scenarios in which the dominant force is transmission limitation and scenarios in which the main force shaping the microbiome is selection that the host environment imposes.

(2)  The Zeng et al. (2017) framework includes microbial influence on host fitness which is slightly different from this influence in our framework: their approach focuses on a functional perspective, and each microbe can contribute to one or more of certain functions that benefit or harm the host. Our framework is structured in a modular way that allows such exploration in future studies, but in the version implemented here we explore—as describe above—several schemes of microbial contribution to its host that make very minimal assumptions about the functional profile of the microbiome. We compare different fitness schemes in which each microbe has an additive contribution to its host's fitness, each sampled from a certain distribution, showing that different distributions would lead to qualitatively different selection dynamics.

### Reporting summary
Further information on research design is available in the Nature Portfolio Reporting Summary linked to this article.

### Data availability
Representative samples of the raw data as it was generated during runs of the simulations described in this paper is available in github.com/itaydaybog/MicrobiomeFramework. The full code is provided, allowing the reproduction of all data and figures used in this study. All produced data can be received from the authors upon request.

### Code availability
The code of the framework described in this paper, alongside code for the results presentation and analysis, is provided in github.com/itay-daybog/MicrobiomeFramework[89]. The model was implemented in Python 3.7.

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

## Acknowledgements
We thank Tommy Kaplan, Amir Bar, Marcus Feldman, and members of the Kolodny lab for insightful comments and discussions. O.K. and I.D. were funded by the Israel Science Foundation (ISF; 1826/20), the United States – Israel Binational Science Foundation (BSF), and the Gordon and Betty Moore Foundation.

## Author contributions
O.K. conceived the framework in this project. I.D. and O.K. designed the study. I.D. implemented the model, derived the results, and analyzed them. I.D. and O.K. wrote the paper.

## Competing interests
The authors declare no competing interests.
