## [Peer Review File · Nature Communications]

Solutions to the microbiome diversity conundrum wherein the microbiome determines host fitness but differs among individualsReviewers' Comments:

Reviewer #1:

Remarks to the Author:

This study presents a theoretical framework that enables hypothesis testing regarding microbiome-host fitness dynamics. The authors present this framework as a way to understand what they termed "the microbial β -diversity conundrum" – microbiomes are known to influence host fitness yet vary dramatically between fit individuals. The most notable finding from their approach was that the law of large numbers explains this conundrum well in that there is low variance in host fitness when there is a large number of species present in the microbiome thus preventing selection from acting on any fitness benefits of the microbiome. In this regard, the results of the paper support the authors main conclusions and claims. Although the model the present makes various assumptions that are not widely biologically applicable (i.e. asexual host reproduction), there is a clear benefit of this oversimplified model to tease apart factors of the microbiome that impact the host population that could be of interest to the larger microbiome research community. The authors also show various ways in which the role of the microbiome on host populations can vary widely despite the influence of the microbiome to an individual's fitness using this model. I do think some more justification of the parameterization of their model could be helpful, particularly with respect to how the high- and low-species microbiomes were created (Supp. Tables S1a,b). i.e. how did the authors choose the values for carrying capacity of microbial pops and microbiome as a whole? Otherwise, I think the authors do a good job of presenting the potential use and benefit of this null model and assuming their code is made publicly available, the work should be reproduceable and available for other researchers to use in their own inquiries.

Other minor comments:

Again, more justification on parameters would be helpful: How/why did you choose $N=2000$ as your number of microbial taxa? What happens to the model if this number is made smaller/larger?

Why did you choose to use the species-rich configuration when comparing host population size? You already see minimal influence of microbiome on host fitness in this scenario. Would your conclusion that in smaller populations the microbiome is limited in its ability to influence host selection? Did you test to see if it would have a larger effect in small pop if less diverse microbiome?

Fig S5a,b. Suggest moving to main text. It would be helpful to see these side by side with the vertical transmission (Fig.5)

Fig 9 legend: Should also mention this was done with a species-rich microbiome

Reviewer #2:

Remarks to the Author:

- What are the noteworthy results?

The paper investigates three scenarios on how the microbiota can affect a host population in the evolutionary sense. I liked how each of the scenarios were laid out with specific details about the fitness scores, the evolutionary aspect of the populations and the comparisons provided.

- Will the work be of significance to the field and related fields? How does it compare to the established literature? If the work is not original, please provide relevant references.

The work will be significant to the field as it gives a working agent-based model which provides insight into the interactions between the microbiome and the host. I also like how the model can be modified to answer future questions.

- Is there enough detail provided in the methods for the work to be reproduced?

The methods section is detailed enough for reproduction. As the code is not available at present, it is

difficult to say if the model can be easily implemented by everyone.
I have no other comments about the paper.

Reviewer #3:

Remarks to the Author:

The authors use a computational agent-based framework to address what they refer to as the microbial β -diversity conundrum, specifically, the apparent paradox that whereas microbiomes appear to provide significant functions for the host and its survival and thus contribute to host fitness, there is high variation in microbiome composition between hosts (i.e., high β -diversity). If selection is operating, the authors argue, then we expect to see the same beneficial microbial taxa in different hosts, thus reducing β -diversity.

My initial concern is whether there really is a conundrum in the first place. When we think of selection (particularly directional or adaptive selection) as reducing diversity in a population, we typically think of this selection acting on different phenotypes of the same trait, e.g., resistance to pathogens, susceptibility to sickle-cell anemia, etc. It is not clear that microbiomes can be characterised in the same way. If we use the collection of traits as an analogy for the microbiome, with each OTU performing a particular function (in much the same way that a trait does), then different individuals in a population can have very different combinations of phenotypic traits (and by analogy, different microbes). And of course, I haven't even mentioned diversifying selection which acts to increase trait diversity. It would be really useful if the authors expand on what they mean when they say that there is a conundrum. Perhaps an analogy such as the one I have used will help me (and I expect other readers) to understand this.

The computational framework that the authors have used -- an agent-based forward simulation model with discrete generations of hosts, and microbial acquisition from parents and environment -- is similar to that used by Zeng et al (2015, 2017, cited as refs 33 and 34 in the manuscript), and has been shown to be useful. An area where the authors depart from Zeng et al (2017) is in the assignment of fitnesses to hosts and microbes. The authors' method is, in my opinion, simpler and more efficient, and may be a better way of incorporating selection into the framework. Another point of difference with Zeng et al (2017) is that the authors focus on some specific parameters in the model: the number of microbial species in host microbiomes, and the host population size.

The authors' results indicate that hosts with species-rich microbiomes have high β -diversity but lower fitness variance from host to host. In contrast, species-poor microbiomes have much larger host-to-host fitness variation, although β -diversity is still high. The authors argue that the difference in the size of fitness variances between species-rich and species-poor hosts is due to the law of large numbers, where fitness scores essentially regress more closely to the overall mean with large numbers of microbial species.

Whereas I can understand why the variances of fitness scores is different between species-rich and species-poor hosts, I don't understand why the β -diversity is the same. In fact, in the discussion, the authors say "As such, we expect that species-poor microbiome configurations are more likely to drive selection and thus be more conserved", which appears to contradict the results where β -diversity is high under both scenarios. Am I missing something?

The authors go on to show that time to coalescence is a function of population size when selection is in play: larger host populations coalesce more quickly. If the analogy of microbiome dynamics with population genetics holds, then this is expected -- selection acts more strongly on large populations, leading to more rapid fixation. But here is a question: what happens to β -diversity as host population size changes?

In summary, what the authors have shown is that selection can operate when β -diversity is high, and this depends on the richness of the microbiomes, and host population size. Returning to the point I made initially about the β -diversity conundrum, I remain concerned because it is not clear to me, at least, how we should think about β -diversity, and particularly what we expect it to be. What is baseline β -diversity? How does it vary with the different model parameters? Is it possible to have the same selective outcomes as demonstrated in this paper, if hosts had low β -diversity? If so, has this paper address a conundrum or simply showed how the factors/process they have identified affect the fitnesses of hosts?

Allen Rodrigo

Dear reviewers,

We have found the comments we received constructive and insightful. The manuscript has benefited from them, and we believe it is now much clearer. We believe our responses below will alleviate all concerns raised / clarifications requested. Thank you very much!

We have uploaded a “clean” version of the revised main text, alongside a version with changes tracked. The supplementary material is uploaded as a separate file. Line numbers mentioned below refer to the “clean” version.

COMMENTS

Reviewer #1:

This study presents a theoretical framework that enables hypothesis testing regarding microbiome-host fitness dynamics. The authors present this framework as a way to understand what they termed “the microbial β -diversity conundrum” – microbiomes are known to influence host fitness yet vary dramatically between fit individuals. The most notable finding from their approach was that the law of large numbers explains this conundrum well in that there is low variance in host fitness when there is a large number of species present in the microbiome thus preventing selection from acting on any fitness benefits of the microbiome. In this regard, the results of the paper support the authors main conclusions and claims. Although the model the present makes various assumptions that are not widely biologically applicable (i.e. asexual host reproduction), there is a clear benefit of this oversimplified model to tease apart factors of the microbiome that impact the host population that could be of interest to the larger microbiome research community. The authors also show various ways in which the role of the microbiome on host populations can vary widely despite the influence of the microbiome to an individual’s fitness using this model. I do think some more justification of the parameterization of their model could be helpful, particularly with respect to how the high- and low-species microbiomes were created (Supp. Tables S1a,b). i.e. how did the authors choose the values for carrying capacity of microbial pops and microbiome as a whole? Otherwise, I think the authors do a good job of presenting the potential use and benefit of this null model and assuming their code is made publicly available, the work should be reproduceable and available for other researchers to use in their own inquiries.

Thank you!

The values for the two categories of microbiomes were chosen to be on the order of magnitude of some empirically observed microbiomes and per-species overall population sizes; we now added clarification and some references (see supplementary section 1). The qualitative patterns that we observe are not dependent on the specific details of these choices – to demonstrate this, we now added an exploration of several additional microbiome structures (Fig S3 and respective results, which are qualitatively the same as the ones in the main text, in Figs S8-S9).

The simulation code and code for figure generation have been made public, at <https://github.com/itaydaybog/MicrobiomeFramework> as is now stated in the main text.

A version with updated embedded comments, the specific parameter values necessary to reproduce our results, and examples of data / output files will be made available shortly.

Other minor comments:

Again, more justification on parameters would be helpful: How/why did you choose $N=2000$ as your number of microbial taxa? What happens to the model if this number is made smaller/larger?

This is a fairly arbitrary choice, and the results are not sensitive to it; the reasoning behind it was that there is high environmental diversity that can be sampled (thousands to tens of thousands of species), but that the vast majority of species in the environment are generally extremely limited in their ability to stably colonize a certain host (e.g. human) gut, and can thus be ignored. This is now explained in the supplementary, and we show in the supplementary that our results are robust to replacing the value of N from 2000 to 10,000 or to 1000 (Figs S3, S9-S10, and accompanying text). We've also added a bit more detail regarding the respects in which the microbiome structures we choose resemble realistic ones, with several additional references (lines 252-258).

Why did you choose to use the species-rich configuration when comparing host population size? You already see minimal influence of microbiome on host fitness in this scenario. Would your conclusion that in smaller populations the microbiome is limited in its ability to influence host selection? Did you test to see if it would have a larger effect in small pop if less diverse microbiome?

Thank you - this is now made clearer. As you point out, one prediction that our model offers is that in small populations, the microbiome's ability to influence the dynamics of host selection would be limited if the impact on host fitness does not create large fitness differences between individuals/lineages. However, this is true more broadly than what is highlighted in our model: natural selection is generally non-efficient in small populations with respect to alleles/types that differ in their fitness only slightly (not only in a microbiome context); this is a major tenet of the "nearly neutral theory of evolution", and has been derived analytically in the late '60s and early '70s by Ohta, Kimura, and others. We now point this out and cite several relevant publications (lines 426-430, 440-443). Our motivation to explore different population sizes in this respect was that – given the dependency of natural selection on population size when fitness differences are small – we might find (as we do) that the microbiome's ability to influence host selection dynamics is "saved" when the population of hosts is very large. This is also why we focused on the species-rich configuration in this exploration: in the species-poor scenario there is enough fitness-difference between individuals, due to stochasticity in the microbes' contributions to fitness, to drive an effect on selection dynamics even in relatively small population sizes of hosts. Thus, increasing the population size would make selection due to microbiome differences between lineages even more effective: a further push in the same direction. For the sake of completeness, we now added such an exploration in S10.

Fig S5a,b. Suggest moving to main text. It would be helpful to see these side by side with the vertical transmission (Fig.5)

Done.

Fig 9 legend: Should also mention this was done with a species-rich microbiome

Done.

Reviewer #2 (Remarks to the Author):

- What are the noteworthy results?

The paper investigates three scenarios on how the microbiota can affect a host population in the

evolutionary sense. I liked how each of the scenarios were laid out with specific details about the fitness scores, the evolutionary aspect of the populations and the comparisons provided.

Thank you!

- Will the work be of significance to the field and related fields? How does it compare to the established literature? If the work is not original, please provide relevant references.
The work will be significant to the field as it gives a working agent-based model which provides insight into the interactions between the microbiome and the host. I also like how the model can be modified to answer future questions.

Thank you! We're glad that this point is clear to the reader; we ascribe much importance to it.

- Is there enough detail provided in the methods for the work to be reproduced?
The methods section is detailed enough for reproduction. As the code is not available at present, it is difficult to say if the model can be easily implemented by everyone.

The simulation code and code for figure generation have been made public, at <https://github.com/itaydaybog/MicrobiomeFramework> as is now stated in the main text.

A version with updated embedded comments, the specific parameter values necessary to reproduce our results, and examples of data / output files will be made available shortly.

I have no other comments about the paper.

Reviewer #3 (Remarks to the Author):

The authors use a computational agent-based framework to address what they refer to as the microbial β -diversity conundrum, specifically, the apparent paradox that whereas microbiomes appear to provide significant functions for the host and its survival and thus contribute to host fitness, there is high variation in microbiome composition between hosts (i.e., high β -diversity). If selection is operating, the authors argue, then we expect to see the same beneficial microbial taxa in different hosts, thus reducing β -diversity.

My initial concern is whether there really is a conundrum in the first place. When we think of selection (particularly directional or adaptive selection) as reducing diversity in a population, we typically think of this selection acting on different phenotypes of the same trait, e.g., resistance to pathogens, susceptibility to sickle-cell anemia, etc. It is not clear that microbiomes can be characterised in the same way. If we use the collection of traits as an analogy for the microbiome, with each OTU performing a particular function (in much the same way that a trait does), then different individuals in a population can have very different combinations of phenotypic traits (and by analogy, different microbes). And of course, I haven't even mentioned diversifying selection which acts to increase trait diversity. It would be really useful if the authors expand on what they mean when they say that there is a conundrum. Perhaps an analogy such as the one I have used will help me (and I expect other readers) to understand this.

Thank you. We have added a paragraph that explains more explicitly why one might expect reduced diversity in fitness-influencing traits and that links this idea to such expectations/observations in many biological, and even non-biological, contexts (Lines 49-68). We agree that there are many potential forces that might allow/facilitate the stable maintenance of diversity, and we now mention several

such mechanisms; such cases, however, demand explanation: if natural selection is operating, and all players seem to be playing the same game, why don't they all converge to the best strategy, or get purged out of the game? Possible explanations that come to mind are things like trade-offs between alternative benefits, balancing selection through negatively dependent frequency advantage that strategies might have, or character displacement / niche separation (which are now mentioned in the added paragraph). The point we make, and we hope is now clearer, is that it makes sense to view the standing diversity as a challenge that begs explanation. We firmly believe in the value of null and neutral models (I think we share this sentiment), and we attempt to offer explanations that are in this realm, more parsimonious than the above-mentioned mechanisms.

The computational framework that the authors have used -- an agent-based forward simulation model with discrete generations of hosts, and microbial acquisition from parents and environment -- is similar to that used by Zeng et al (2015, 2017, cited as refs 33 and 34 in the manuscript), and has been shown to be useful. An area where the authors depart from Zeng et al (2017) is in the assignment of fitnesses to hosts and microbes. The authors' method is, in my opinion, simpler and more efficient, and may be a better way of incorporating selection into the framework.

Thank you!

Another point of difference with Zeng et al (2017) is that the authors focus on some specific parameters in the model: the number of microbial species in host microbiomes, and the host population size. The authors' results indicate that hosts with species-rich microbiomes have high β -diversity but lower fitness variance from host to host. In contrast, species-poor microbiomes have much larger host-to-host fitness variation, although β -diversity is still high. The authors argue that the difference in the size of fitness variances between species-rich and species-poor hosts is due to the law of large numbers, where fitness scores essentially regress more closely to the overall mean with large numbers of microbial species. Whereas I can understand why the variances of fitness scores is different between species-rich and species-poor hosts, I don't understand why the β -diversity is the same. In fact, in the discussion, the authors say "As such, we expect that species-poor microbiome configurations are more likely to drive selection and thus be more conserved", which appears to contradict the results where β -diversity is high under both scenarios. Am I missing something?

Thank you for pointing this out; this was not phrased clearly enough, and is now clarified.

Beta-diversity in our simulations changes over time: at the beginning of each simulation, beta-diversity is high in all cases because the composition of each host's microbiome is randomly assigned; this is true for both species-rich and species-poor microbiomes. This is now pointed out (lines 273-280; added figures S11-S12 and their discussion near the end of the supplementary): as time progresses, beta-diversity changes: if a certain composition bestows significant selective benefits to its host, it spreads in the population, and beta-diversity decreases as a result.

Notably, for simplicity, we treat in the main text the scenario in which transmission is purely vertical, and the dynamics are, in a sense, the closest to genetic transmission; we treat horizontal/oblique and mixed selection in the supplementary. (In the current study -- as in Zeng et al 2015 -- we use a neutral model with respect to within-microbiome dynamics, i.e. we do not consider dynamics of selection on microbial species that are separate from the selection dynamics of the host, selection that results from competition among microbial species. We do so because this would introduce complications that make things less tractable and would be confusing in the context of the conundrum we address; we have treated such dynamics elsewhere, in ref 50, in PNAS).

We now point out (lines 273-282) that beta-diversity is initially high in all scenarios, but where selection is effective it rapidly drops, as lineages with better microbiomes are selected. We further elaborate in the last section of the supplementary material.

In the scenarios where selection is not effective the beta-diversity also eventually drops (in the purely-vertical transmission scenario), because drift leads to the fixation of a single host lineage and its microbiome; however, this takes much longer (mentioned now in the discussion of S11-S12 in the supplementary). As a result, the most interesting time points to consider for understanding what beta-diversity should be expected to look like are intermediate timepoints (see newly-added figure S11, and examples of beta-diversity over time in S12).

The authors go on to show that time to coalescence is a function of population size when selection is in play: larger host populations coalesce more quickly. If the analogy of microbiome dynamics with population genetics holds, then this is expected -- selection acts more strongly on large populations, leading to more rapid fixation.

Definitely – this well-known phenomenon (in genetics) led us to explore the parameter of population size in the microbiome context as well. The fact that the observed pattern aligns with what is known in population genetics and stems from the same underpinnings has been highlighted (lines 426-430, 433-435) and several additional references (Kimura, Ohta, and others) have been added.

But here is a question: what happens to β -diversity as host population size changes?

We have now added figure S11 that depicts beta-diversity at several time-points for the species-rich microbiome at population sizes 20, 200, and 2000.

In summary, what the authors have shown is that selection can operate when β -diversity is high, and this depends on the richness of the microbiomes, and host population size. Returning to the point I made initially about the β -diversity conundrum, I remain concerned because it is not clear to me, at least, how we should think about β -diversity, and particularly what we expect it to be. What is baseline β -diversity? How does it vary with the different model parameters? Is it possible to have the same selective outcomes as demonstrated in this paper, if hosts had low β -diversity? If so, has this paper address a conundrum or simply showed how the factors/process they have identified affect the fitnesses of hosts?

These questions helped us realize that more clarification is needed, and such was added via several phrasing-changes and an added paragraph in the discussion section (lines 511-522).

To our best understanding, there is no clear baseline/expectation regarding microbiome expected beta-diversity, not in the real world in general (empirically, it varies hugely between species, host body sites, and contexts; now highlighted in lines 467-470), and not in our model. In our model, beta-diversity is an emergent property that depends on the model parameters from the start (i.e. at generation 1 of the simulation), and changes over time in each simulation as a result of the population dynamics. For generality, we did not attempt to simulate real-life parameters of any particular system in this case, and we accordingly don't have any particular predictions regarding beta-diversity in absolute terms (this is now explicitly clarified in the discussion).

Because both alpha and beta diversity are so dependent on the model parameters, which can take on a huge range of values, we propose that in a computational framework, thinking about beta-diversity is most insightful within the same context. Thus, we compare for example the dynamics that occur with

the species-poor microbiome with those that play out with the species-rich microbiome by making only minimal changes in the parameters of the microbiome assembly, while keeping all other parameter values constant.

It is definitely possible to find in our framework settings in which selection occurs and leads to low beta-diversity (or is correlated with it). This occurs, for example, in the scenario of species-poor populations, as noted above, and in the scenario of large populations with species-rich microbiomes. We have added figures S11 – S12 which depict such scenarios.

Allen Rodrigo

Thank you for the thoughtful comments and suggestions!

Reviewers' Comments:

Reviewer #1:

Remarks to the Author:

The author's have done an excellent job at incorporating all of the reviewer's comments into the revised manuscript. I have no additional concerns and support publication of the current version.

Reviewer #3:

Remarks to the Author:

I am satisfied that the authors have addressed the reviewers' comments.

I do have one final suggestion: The authors appear to claim that the computational framework that they have adopted is their own. They say, for instance:

"Our framework can also be used to explore scenarios in which β -diversity is low in the first place due to the assembly process or other constraints, or to study cases in which fitness is only partially dependent on the microbiome's composition."

I am having difficulty seeing where their framework is more than just an extension to the framework developed by Zeng et al. I note that the authors do not cite Zeng et al in the Methods section, nor do they acknowledge the similarities between the two frameworks.

Of course, as the senior author on Zeng et al's papers, I readily admit having an interest in seeing that our work is properly acknowledged. However, it is possible that I am wrong, and the authors have developed a model that is substantially different from that developed by Zeng et al, and I cannot see the difference; in which case, it is useful to point out what these significant differences are.

Allen Rodrigo

Response to reviewers' comments

Reviewer #1 (Remarks to the Author):

The author's have done an excellent job at incorporating all of the reviewer's comments into the revised manuscript. I have no additional concerns and support publication of the current version.

Thank you!

Reviewer #3 (Remarks to the Author):

I am satisfied that the authors have addressed the reviewers' comments.

Thank you!

I do have one final suggestion: The authors appear to claim that the computational framework that they have adopted is their own. They say, for instance:

"**Our framework** can also be used to explore scenarios in which β -diversity is low in the first place due to the assembly process or other constraints, or to study cases in which fitness is only partially dependent on the microbiome's composition."

I am having difficulty seeing where their framework is more than just an extension to the framework developed by Zeng et al. I note that the authors do not cite Zeng et al in the Methods section, nor do they acknowledge the similarities between the two frameworks.

Of course, as the senior author on Zeng et al's papers, I readily admit having an interest in seeing that our work is properly acknowledged. However, it is possible that I am wrong, and the authors have developed a model that is substantially different from that developed by Zeng et al, and I cannot see the difference; in which case, it is useful to point out what these significant differences are.

Allen Rodrigo

Thank you for this comment; we have now noted similarities between our framework and previous ones in the beginning of the results section, where the model is first presented, and added a fairly detailed discussions of similarities and differences at the end of the Methods section. We have attached a "clean" version of the main text, alongside one in which these changes are tracked.